# Mangrove-Derived Endophytic Bacteria Enhance Growth, Yield, and Stress Resilience in Rice

**DOI:** 10.3390/ijms26199317

**Published:** 2025-09-24

**Authors:** Amal Khalaf Alghamdi, Anamika Rawat, Waad Alzayed, Sabiha Parween, Arun Prasanna Nagarajan, Maged M. Saad, Heribert Hirt

**Affiliations:** 1D21 Desert Research Initiative, Biological and Environmental Science and Engineering Division, King Abdullah University of Sciences and Technology, Thuwal 23955, Saudi Arabia; amal.alghamdi@kaust.edu.sa (A.K.A.); anamika.rawat@kaust.edu.sa (A.R.); waad.alzayed@kaust.edu.sa (W.A.); sabiha.parween@kaust.edu.sa (S.P.); arun.nagarajan@kaust.edu.sa (A.P.N.); maged.saad@kaust.edu.sa (M.M.S.); 2Department of Botany and Microbiology, College of Science, King Saud University, Riyadh 11451, Saudi Arabia

**Keywords:** waterlogging, salt stress, microbiome, rice, Arabidopsis, sustainable agriculture

## Abstract

Global climate change increasingly challenges agriculture with flooding and salinity. Among strategies to enhance crop resilience to these stresses, we tested several endophytic bacterial strains from mangroves, which are permanently exposed to flooding and high salinity. We show several strains that can enhance flooding and salinity tolerance in Arabidopsis and rice plants. Two strains and their combination massively enhanced the growth and yield of *Oryza sativa* cv. Nipponbare under both soil and hydroponic growth conditions with and without salt treatment. The bacteria-induced transcriptome changes in *O. sativa* roots, particularly related to ABA-signaling and lignin and suberin deposition in root tissues, explain the altered responses of colonized rice plants to hypoxic and saline stress conditions. Importantly, bacterially colonized rice plants exhibited enhanced yield and improved grain quality. These results show that microbes can be a powerful tool for enhancing the yield and resilience of rice to hypoxic and saline stress conditions.

## 1. Introduction

Due to global warming, agriculture is challenged by the rising incidence of adverse weather events. Extreme events that alter water availability, like droughts and floods, threaten food security [1]. On a global scale, floods caused almost two-thirds of all damage and loss to crops between 2006 and 2016, costing billions of dollars [2].

Excessive water supply induces hypoxia in plants [3], increases the vulnerability to pathogen attack [4], and limits light flow to the plant [5]. During recovery after flooding, plants experience oxidative stress [6] and must remobilize nutrients to achieve a normal homeostatic state [7]. In recent years, flooding stress and its derivatives, like submergence, waterlogging, hypoxia, and anoxia, were investigated extensively in plants to identify molecular elements that may play a role in tolerance to flooding [8]. Plants respond to flooding and the associated stress by gene expression changes regulated at multiple levels, including transcription [3,9] translation [10] and epigenetics [11].

Salinity is another global problem for plants that is becoming increasingly important. Soil salinity inhibits plant growth and development, strongly reducing crop yield. Salt stress induces both osmotic changes and toxicity in plants, resulting in reduced water availability and sodium accumulation [12]. Halophytes are plants adapted to saline soils using specialized strategies to deal with salinity. Since most crop species are salt-sensitive, efforts are underway to make them more salt-tolerant to maintain crop yield and potentially use saline irrigation.

The use of harmful chemical fertilizers and pesticides has negative impacts on the environment and human health. This situation has generated increasing interest in using beneficial microorganisms to develop sustainable agri-food systems [13]. Recent research showed the remarkable importance and range of microbial applications for enhancing the growth and health of plants, and plant growth-promoting bacteria (PGPBs) rapidly developed as a means to enhance tolerance to abiotic and biotic stresses [14]. So far, however, no microbes have been characterized that can enhance flooding tolerance and ensure the productivity of crops.

Recent studies increasingly demonstrate that PGPBs isolated from halophytes, such as mangroves, can enhance stress tolerance in plants. Yang et al. [15] investigated the diversity and functional roles of phyllosphere microbial communities in mangrove plants and reported that two strains, *Pantoea stewartii A* and *Bacillus marisflavi Y25*, improved salt tolerance in rice by modulating osmolytes, including sugars and proline, and regulating salt-stress-related genes such as *OsSOS1*, *OsPIN1*, and *OsCIPK15*. Similarly, the endophytic bacterium *Delftia tsuruhatensis DYX29*, isolated from *Kandelia obovata*, enhanced rice growth under salt stress by promoting osmoregulation, increasing antioxidant enzyme activity, maintaining ion homeostasis, and modulating phytohormone levels [16].

PGPRs enhance plant stress resilience through multiple mechanisms. One such mechanism is through the production of exopolysaccharides, which helps plants by improving soil aggregation, water retention, nutrient availability, root adhesion and biofilm formation, as well as stress tolerance by sequestering toxic ions under saline conditions [17,18,19]. PGPRs also modulate the phytohormone levels to alleviate stress in plants. The ACC deaminase-producing PGPRs break down ACC, an ethylene precursor, reducing stress-induced ethylene levels and mitigating growth inhibition [20,21]. Ethylene also regulates ROS accumulation under salt stress via the receptor-like kinase SIT1 [22]. By lowering ethylene, ACC deaminase-active PGPRs increase antioxidant enzyme activities (SOD, CAT, POD), protecting cells from oxidative damage [19]. These mechanisms underscore the interplay between ethylene and ROS and how PGPR alleviates stress to improve plant performance.

Despite these advances, no evidence shows that bacteria can be used to enhance flooding tolerance and ensure the productivity of crop plants. Therefore, this study aimed to investigate the role of mangrove-derived bacteria in supporting plants under these conditions. Specifically, we sought to (i) test whether mangrove bacteria enhance the resilience of *Arabidopsis thaliana* to salinity under waterlogging conditions, (ii) determine their functional role in enhancing the hypoxia resilience of *O. sativa* cv. Nipponbare using hydroponics, (iii) assess the influence of selected strains on rice growth under hydroponic normal and salt stress conditions, as well as the soil performance and yield. By addressing these objectives, this study provides novel insights into the transcriptomic regulation and root structure changes underlying the better performance of the mangrove-bacteria-treated plants compared to the control and their potential applications in sustainable agriculture.

In this work, we used bacteria derived from the mangrove *Avicennia marina,* isolated from the shore of the Red Sea, whose roots are constantly submerged but also undergo repetitive flooding at the whole plant level. In addition, *A. marina* has adapted to grow under high-saline conditions. We, therefore, tested several *A. marina* microbes first on *Arabidopsis thaliana* under waterlogging and then under waterlogging saline conditions. Selected single and multiple strains were then tested on *Oryza sativa cv.* Nipponbare under various conditions. Overall, our results identified strains that can protect rice plants from root submergence and saline stress. In addition, we could massively reduce the life cycle of rice, potentially allowing a shift from two to three harvests per year under optimal conditions.

## 2. Results

### 2.1. Mangrove Bacteria Enhance the Resilience of Arabidopsis thaliana to Salinity Under Waterlogging Conditions

To mimic the hypoxic conditions naturally occurring in the mangrove ecosystem, *A. thaliana* seedlings were inoculated with bacterial strains and grown by a waterlogging method (Appendix A). Based on previous analyses (unpublished data), 16 isolates were tested for their ability to enhance plant growth on ½ MS in waterlogging growth conditions. When compared to non-inoculated controls, five strains (AK031, AK116, AK164, AK171, AK225) showed significant enhancement (35–107%) of plant growth under both waterlogging (1/2MS) and saline waterlogging conditions (1/2MS + 100 mM NaCl) (Table 1, Figure 1).

### 2.2. Mangrove Bacteria Enhance the Resilience of O. sativa cv. Nipponbare Using Hydroponics

To test for the beneficial effects of the *A. marina bacterial* strains on rice, we tested eight selected strains on rice grown in hydroponic as well as hydroponic saline conditions. Among the eight tested strains (AK031, AK073, AK116, AK157, AK164 [23], AK171 [24], AK181, AK255), several exhibited notable growth enhancements in inoculated rice plants compared to the non-inoculated plants under the hydroponic conditions (Figure 2A–C; Appendix A). Under saline hydroponic conditions (100 mM NaCl), two strains (AK164 and AK171) not only promoted growth (Figure 2D,E) but also significantly increased the number of tillers per plant in inoculated rice relative to non-inoculated plants (Figure 2F; Appendix A).

### 2.3. AK164, AK171, and BiCom Enhance Rice Growth Under Hydroponic Normal and Salt Stress Conditions

We recently published the whole genome sequences of AK164 and AK171, and have shown that both these bacterial strains possess genes associated with PGP traits [23,24]. AK164 and AK171 were next analyzed alone and in combination (BiCom) for their potential to enhance the growth and salt stress resistance of rice. As described beforehand, we used a hydroponic system to assess various growth enhancement parameters in hydroponic normal conditions (Figure 3A–D, Appendix A). To evaluate efficacy, we assessed various growth parameters, including dry weight (DW) and the number of tillers, compared to non-inoculated plants (MOCK).

Both AK164 and AK171 significantly increased the shoot DW of inoculated rice plants by 42.5% and 49.7%, the root DW by 64.2% and 112.6%, and the number of tillers by 35.3% and 82.3%, respectively, when compared to mock-inoculated plants (Figure 3A–D, Appendix A). Subsequently, we investigated the combined effects of these two strains as a BiCom, which resulted in a considerably higher DW in both shoots and roots when compared to the single inoculated strains, resulting in 63.8% and 118.6% increase, respectively, compared to mock-inoculated plants (Figure 3B,C; Appendix A). Importantly, the tiller number per plant for BiCom inoculated rice increased from 35.3% (AK164) and 82.3% (AK171) to 97% (AK164 + AK171), as shown in Figure 3D.

Under hydroponic saline conditions (100 mM NaCl) (Figure 3E–H), rice plants inoculated with AK164, AK171, or BiCom exhibited significant enhancements in growth compared to the mock (Figure 3E; Appendix A). Notably, the BiCom inoculation showed substantial enhancement in dry weight in both shoots (up to 120%) and roots (up to 110%). Also, the number of tillers per plant was higher (up to 79%) than the mock (Appendix A). These findings underscore the potential of AK164 and AK171, both individually and in combination, to promote plant growth and mitigate the effects of abiotic stress.

### 2.4. AK164 and AK171 Enhance Soil-Grown Rice Performance and Yield

We then conducted greenhouse experiments (Appendix A) to evaluate the growth of rice seeds inoculated with individual bacterial strains (AK164 and AK171) and their BiCom with/without salinity treatments. After 90 days of growth, we analyzed various agronomic parameters to assess the effects of bacterial inoculation on rice growth (Figure 4 and Figure 5, Appendix A). Under non-stressed conditions (0 mM NaCl) as shown in Figure 4A, we observed that AK164 or AK171 alone already substantially improved most agronomic parameters (Figure 4B–I; Appendix A). Significant increases were observed in the production of the number of tillers/plant (31–66%), panicles/plant (43–104%), and spikelet/panicle (15–26%). As before, BiCom-treated rice showed superior PGP metrics in all these categories. Most importantly for yield assessment, the number and weight of grains per plant also showed increases of 48% and 29% for AK164, 33% and 25% for AK171, and 69% and 54% for BiCom-treated rice plants, respectively (Figure 4E,F; Appendix A). Furthermore, both panicles and spikelet were clearly longer, which explains the significant enhancement of the weight of panicles per plant in the plants inoculated with the BiCom compared to non-inoculated (mock) plants (27% and 31%), as shown in Figure 4G. In addition, the size and length of seeds (Figure 4H,I) were significantly enhanced, indicating the potential of bacterial inoculation to improve seed quality and yield in rice cultivation.

Under saline conditions (100 mM NaCl), the plants treated with either of the bacteria AK164 or AK171 alone enhanced the agronomic parameters (Figure 5A). Still, BiCom notably showed a further increase in the number of tillers by 90% (Figure 5B, Appendix A), panicles per plant by 550% (Figure 5C), spikelet per panicle by 40% (Figure 5D), as well as the number and weight of grains per plant (Figure 5E,F; Appendix A).

The yield of AK164, AK171, or BiCom inoculated rice was remarkably enhanced as the weight and number of grains per plant were significantly increased by 324 and 284%, respectively (Figure 5E,F; Appendix A). Panicles and spikelets were clearly longer and more filled with seeds, which explains the significant enhancement of the weight of panicles per plant in rice inoculated with the bacterial strains when compared to non-inoculated plants (1032% and 4915%), as shown in Figure 5G (Appendix A). In addition, the size and length of seeds (Figure 5H,I) were significantly enhanced, indicating the potential of the bacteria to improve seed quality and yield in rice cultivation under saline conditions.

### 2.5. AK164 and AK171 Induce Upregulation of Rice ABA and Carotenoid Signaling Pathways

To gain a better understanding of the molecular mechanisms by which these bacteria enhance the growth of rice plants, we performed RNA seq analysis of AK164 and AK171 colonized rice plants with and without salt stress conditions. Since the mangrove bacterial strains are both root epiphytes, we conducted this analysis separately for rice root and shoot tissues and concentrated on the rice root transcriptome (Figure 6A–C). For roots of AK164-colonized rice plants, 71 genes were found to be up-regulated (Figure 6B), among which the genes for the two carotenoid biosynthesis enzymes CCD1 and CCD8a were the SAPK9 protein kinase, the NRT1/AIT1 transporter, and the histone demethylase JMJ30 (Figure 6C). For the 12 down-regulated genes (Figure 6B), the only GO term was found for transporters, but almost exclusively contained aquaporin genes PIP1;2, PIP1;3, PIP2;A13, PIP2;3, PIP2;4, PIP2;5 and TIP2;1 (Figure 6C). In roots of AK171-colonized plants, 107 genes were found to be up-regulated (Figure 6B), comprising again JMJ30, SAPK9, PCL1, and NRT1, but also CCD1 and LOX1, LOX3 (Figure 6C). Interestingly, in the 37 down-regulated genes, the only GO term with high significance (5 × 10^−5^) was retrieved for transporters, most of which comprised the same set of aquaporin genes as found in the AK164-inoculated plants’ transcriptome, except for PIP1;2 (Figure 6C). Although roots of non-colonized plants showed the regulation of massive sets of DEGs under saline conditions, AK164- or AK171-colonized plant roots showed no significant up or down-regulated genes (Figure 6B).

For shoots of AK164-colonized rice plants grown under normal hydroponic conditions, no GO terms were retrieved, neither for up nor down-regulated genes (Appendix A). However, the upregulated shoot gene set in both AK171- and AK164-colonized plants contained the genes for JMJ30, SAPK9, NRT1, the transcription factor PCL1, and the receptor kinase Feronia in control conditions. For the down-regulated 171 genes, no GO terms were obtained. Under saline conditions, the shoot transcriptome of AK164-colonized plants only showed 12 up- and 69 down-regulated DEGs with no significant GO terms. However, SAPK9, NRT1, PCL1, and JMJ30 were again found in the significantly upregulated list of genes.

The high overlap between the root transcriptomes of AK164- and AK171-colonized rice plants was surprising and indicated that the two taxonomically different strains were targeting similar set of genes. Interestingly, a closer look at the genes revealed that most of these differentially expressed genes were either biosynthesis, signaling, or target genes of the carotenoid pathway (Figure 6C). Intriguingly, some of the target genes, such as JMJ30, NRT1, and SAPK9, have been shown to be ABA-regulated. In contrast, the carotenoid biosynthesis genes CCD1 and CCD8 encode oxygenases, producing apocarotenoids including β-cyclocitral and strigolactone.

### 2.6. AK164 and AK171 Promote the Deposition of Secondary Cell Wall Components in Colonized Rice

Aiming to visualize the secondary cell wall components in rice roots, we performed triple staining of rice roots grown hydroponically under normal conditions. Basic Fuchsin has been shown to stain lignin. In contrast, Fluorol Yellow stains suberin. The two dyes were combined to visualize the deposition of these two important components of the cell wall, which play important roles as a barrier in preventing loss of water and solutes and act to enhance plant cell wall rigidity. Rice tolerates hypoxia in the surrounding solution by forming lysigenous aerenchyma and a physical barrier to prevent radial oxygen loss (Figure 7). We observed changes in sclerenchyma layer formation in colonized plants. The sclerenchyma layer is a single-cell file in WT rice roots. While colonized plants showed multiple sclerenchyma layers, AK171 induced multi-serrate sclerenchyma layer formation, which was also seen in the AK164 + AK171 inoculated plants. Multi-serrate sclerenchyma is associated with a greater root lignin concentration and greater tensile strength for plants to survive in low-oxygen conditions. Deposition of suberin in endodermal cells is another adaptation in rice roots. We observed enhanced suberized endodermal cells in rice roots when colonized with PGPB candidates, suggesting enhanced barrier formation to allow water and dissolved nutrients to selectively enter the stele for translocation to the shoot, while preventing the radial oxygen loss from roots into anaerobic soil. In rice plants grown in hydroponic saline conditions, we observed the collapse of root structure in non-colonized plants that were absent in AK164-, AK171-, or BiCom- colonized plants (Figure 8). Since enhanced sclerenchyma, lignin, and suberin deposition can strengthen organ structure, the enhanced robustness of the roots of colonized rice plants may be a reasonable explanation for this finding.

## 3. Discussion

Given the massive global challenges for food security in the coming decades, securing yield is of primary importance in agriculture. Among these challenges, weather pattern changes result in drought, heat waves, and flooding. Whereas flooding of plants induces hypoxia that can result in damage and death in some species, other plant species can resist hypoxia and survive or even thrive under these conditions. The genetic basis of this plasticity has been well studied, and several hormonal pathways and genes have been identified that convey flooding resilience [6,25,26,27,28]. However, little is known about whether plant-associated microbes can enhance flooding resilience in sensitive plant species. We show that several bacterial endophytes derived from constantly flooding-challenged mangroves can enhance flooding tolerance in Arabidopsis and rice. Since mangroves thrive in seawater, we also tested whether these microbes could enhance the salt tolerance of glycophytes. The strains that enhanced flooding tolerance also enhanced salt tolerance in Arabidopsis and rice.

Sixteen strains were analyzed for their capacity to enhance Arabidopsis plant growth on ½ MS plates under standard conditions. To mimic the hypoxic conditions naturally occurring in the natural habitat of the mangrove ecosystem, *A. thaliana* seedlings were then grown using a submerged block method. Among these, AK164 and AK171 were selected as the best performers of rice-inoculated seedlings under normal and saline conditions. *Isoptericola* sp. AK164 enhances the growth of inoculated rice seedlings compared to non-inoculated plants by significantly increasing the shoot DW, root DW, and number of tillers per plant in normal hydroponic conditions. In saline hydroponic conditions, the same parameters were significantly increased compared to mock-inoculated plants. KLBMP 4942 (JX993798) enhanced *Limonium sinense* seed germination by 85% under normal and 43% under saline conditions while significantly increasing the leaf area under saline conditions, potentially through ACC deaminase production and flavonoid accumulation by the beneficial microbes [29].

To test the ability of AK164 and AK171 to improve plant performance synergistically, we generated a BiCom inoculum of the AK164 and AK171 bacterial strains. Compared to non-inoculated rice plants, we found that BiCom enhanced waterlogging tolerance, yield performance, biomass, and the number of tillers. Under hydroponic saline conditions, BiCom enhanced salt tolerance and yield in shoot DW, root DW, and the number of tillers, which revealed an additive beneficial influence on plant growth under normal hydroponic and hydroponic saline conditions. The additional beneficial effects might be due to complementing each other’s ability to form biofilms, produce secondary metabolites, or stabilize colonization of the host plants [30].

In addition, compared to non-colonized rice plants, agronomic and yield parameters significantly increased with either of the strains but to higher levels for the BiCom under both normal and saline soil in greenhouse conditions. Importantly, the life cycle of rice plants was found to be significantly shortened by colonization of either of the two strains, particularly by the BiCom. Our work confirms other studies showing that applying a BiCom enhances plant growth and yield under greenhouse conditions and in the field [31,32].

Although it is very important to identify beneficial microbial strains, it is even more important to understand their modus operandi. We tried to find a conserved pattern in analyzing the rice gene expression patterns in response to two taxonomically different mangrove microbes. Since mangrove strains induced rice tolerance to flooding and salt stress as well as enhanced the yield, we reasoned that the two strains might act by targeting the same signaling pathways in rice plants. The comparative transcriptome analysis of colonized to non-colonized rice roots under hydroponic and hydroponic saline conditions suggested an essential role of several genes (JMJ30, NRT1/AIT, SAPK9, and CCD1). JMJ30, a key epigenetic regulator of histone H3 K27 methylation, was identified as the most highly upregulated common gene. In Arabidopsis, JMJ30 is the key determinant of circadian rhythm. It regulates the flowering time cooperatively with a central oscillator, circadian clock associated1 [33], and late elongated hypocotyl [33,34], thereby modulating the expression of certain regulators involved in abiotic stress [35]. In Arabidopsis, the circadian clock regulates various processes such as gene expression, photoperiodic flowering, and leaf movement [36]. JMJ30 promotes callus proliferation by binding to and activating a subset of *LATERAL ORGAN BOUNDARIES DOMAIN* (*LBD*) genes and primarily demethylates trimethylation at lysine 9 of histone H3 (H3K9me3) at *LBD16* and *LBD29*, rather than H3K27me3 [37]. Nitrogen (N) is the most limiting nutrient for crops. Waterlogging causes losses in soil N through denitrification and nitrate leaching, thereby reducing the N pool available for plants and compromising plant growth. In the same context, the nitrogen transporter gene (NRT1) was upregulated in the roots of AK164- and AK171-inoculated plants under hydroponic normal and saline conditions, resulting in a significant increase in plant biomass. NRT1 is activated by the flowering transcription factor N-mediated heading date-1 (Nhd1), which regulates flowering time. It was also found that Nhd regulates nitrogen uptake and the root biomass in the field [38]. The Stress-Activated Protein Kinase SAPK9 was also upregulated in inoculated plants under hydroponic normal and saline conditions. SAPK9 is a member of the SnRK2 family of protein kinases and one of the core components of the ABA signaling pathway. SAPK9 improves drought tolerance and grain yield in rice by modulating cellular osmotic potential, stomatal closure, and stress-responsive gene expression [39]. SAPK9 functions by phosphorylating ABA-responsive element binding protein (AREB) transcriptional factors, including bZIPs (basic region-leucine zipper), to participate in several biological processes. Interestingly, SAPK9 binds to bZIP77/OsFD1, a key rice flowering regulator [40], and also interacts with OsMADS23, a positive regulator of osmotic stress, to transcriptionally activate *OsNCED2, OsNCED3, OsNCED4,* and *OsP5CR,* which are key components for ABA and proline biosynthesis, respectively. SAPK9 phosphorylation increases the stability and transcriptional activity of OsMADS23, thereby regulating ABA biosynthesis and providing a novel strategy to improve salinity and drought tolerance in rice [41]. In rice, ABA regulates suberin deposition via *TREHALOSE-PHOSPHATE-SYNTHASE* (*OsTPS8*) [42]. Over-expression of *OsTPS8* enhances salinity tolerance without yield penalty.

In contrast, the *ostps8* mutant showed significantly reduced soluble sugars, reduction in Casparian bands, as well as suberin deposition in the roots compared to the WT and overexpression lines [42]. This might explain the suberin deposition in the cell walls of the root cells in the PGPB-inoculated plants in our study (Figure 7 and Figure 8). ABA can also regulate the lignin biosynthesis by phosphorylating the master lignin transcription factor NST1—a substrate for ABA-dependent SnRK2 kinases, suggesting a tight control of ABA signaling in secondary cell wall deposition [43].

Under hydroponic conditions, the roots of inoculated plants upregulated carotenoid cleavage dioxygenases (*CCD1*). CCDs are important biosynthetic enzymes for distinct carotenoids. A ccd1 ccd4 mutant showed fewer meristematic cell divisions and less lateral root branching compared to mock roots [44], which can be rescued by root indigenous β-cyclocitral. In addition, *OsCCD4a*-RNAi-*7* lines showed increases in the leaf carotenoids α-carotene, β-carotene, lutein, antheraxanthin, and violaxanthin relative to non-transgenic plants. Interestingly, seeds of *OsCCD1*-RNAi-*8* plants displayed a 1.4-fold increase in total carotenoids [45]. In contrast, inoculated plant roots exhibited upregulation of the CAROTENOID CLEAVAGE DIOXYGENASE 8 (CCD8) gene. This gene plays a role in the biosynthesis of the phytohormone strigolactone, which enhances tomato yield production [46]. *OsCCD8b* regulates rice tillering, whereas *ZmCCD8* plays essential roles in root and shoot development, with smaller roots, shorter internodes, and longer tassels in *Zmccd8* mutant plants [47].

Abiotic stress, including flooding and salinity, drastically inhibits plant growth and severely reduces productivity [48,49]. Multiple mechanisms are used to restore cellular homeostasis and promote survival [50]. Osmotic stresses, including salt, drought, and/or PEG treatment, have been shown to affect plant cell water balance. For example, the root water uptake capacity (i.e., root hydraulic conductivity) was inhibited by salinity [51,52]. Transcriptome analysis of the roots of rice plants under both hydroponic and saline conditions showed several aquaporins (AQPs) to be downregulated upon bacterial inoculation using AK164 and AK171. The aquaporins are membrane proteins of plasma membranes (PIPs) or the tonoplast (TIPs). Aquaporins are also thought to be involved in stress-coping mechanisms, such as altering the hydraulic conductivity of tissues [53,54]. In rice (*Oryza sativa* L. cv. Nipponbare), 33 aquaporins were identified, including 11 PIPs, 10 TIPs, 10 NIPs, and 2 SIPs [55]. Healthy plants must have a constant water supply is important for growing tissues to maintain turgor pressure upon cell enlargement [56]. AK164- and AK171-inoculated plants showed downregulation of a few genes, of which the AQPs represented the only family of related functions. The concreted regulation of PIP1;2, PIP2;3, PIP2–4, and PIP2;5, suggests a regulation of the plant water status by the two microbial strains. Various studies suggest that a low abundance of AQP proteins reduces water permeability, and a high abundance (over-expression) increases the hydraulic conductivity of biological membranes [57]. It was found that plants with mid- to low-level overexpression of *OsPIP1;1* showed salt resistance [58], implying that OsPIP1;1 plays an important role in regulating water homeostasis in planta. Although no significant changes in PIP1;1 expression were observed in our study under salinity, we observed significant changes in *PIP1;2* under normal conditions of the PGPB-inoculated roots. The RNAi plants of *OsPIP2;1* have decreased root length, surface area, root volume, and root tip number [59]. Additionally, *PIP1;1* has high water channel activity when co-expressed with *PIP2*, and PIP1–PIP2 random hetero-tetramerization not only allows PIP1;1 to arrive at the plasma membrane but also results in enhanced activity of PIP2;1 [60]. These results suggest the importance of both PIP1 and PIP2 aquaporins for controlling water movement across the plasma membrane [61]. In agreement with our findings, the overexpression of *OsPIP1;1* at very high levels markedly decreases fertility, as we found in MOCK plants in normal conditions, whereas gene expression at low or medium levels raises seed yield but does not affect single-grain weight [58]. This suggests a role of OsPIP1;1 in seed set, probably by affecting pollen germination in the stigma or pollen tube growth [62]. Several authors reported that *PIP* expression responded to salt and drought stress in Arabidopsis and rice [63,64,65]. The expression pattern of a particular aquaporin gene may vary within or across plant species. This variability can be rooted in the different cultivars, developmental stages, or stress conditions. In all, it has been stated that TIPs and PIPs together may help the plant adjust water transport through transcriptional regulations to tune the water balance under stress [66,67]. In Arabidopsis and rice, the expression response to dehydration and salinity in some *PIP*s was mediated by ABA [65,68,69]. These data indicated that, as a stress-signaling molecule, ABA is involved in regulating expression of some *TIP* and *PIP* genes in response to abiotic stresses. It was also reported that the expression of rice *TIP*s was significantly upregulated by dehydration, salinity, and ABA treatments. It was found that TIP1 expression in the shoots and roots of rice seedlings was influenced by water stress, salt stress, and exogenous ABA [70]. In addition to water transport, AQPs may contribute to hypoxia tolerance by transporting O_2_ [56], H_2_O_2_, and lactic acid. The complex responses toward oxygen deprivation may involve ethylene, abscisic acid, or other hormonal factors and signaling molecules [71].

## 4. Methods

### 4.1. Bacterial Culture Conditions

The bacteria used in this study, including *Isoptericola* sp. AK164 [23] and *Tritonibacter mobilis* AK171 [24], were isolated from the rhizosphere of *Avicennia marina* seedlings growing on the Red Sea shore in 2019. They have been routinely grown in Zobell marine agar and incubated aerobically at 30 °C. The seeds were inoculated with either the two bacterial candidates individually (AK164 and AK171) or as a combination to create a synthetic community (BiCom). The inoculum of each bacterial strain used for successive experiments was 100 µL of O.D. 0.2 (~10^9^ CFU mL^−1^ using a spectrophotometer at 600 nm). For the BiCom, a mixture of each strain was mixed as 50 μL of each (1:1).

### 4.2. Plant Material and Growth Conditions

The five-day-old *A. thaliana* seedlings (18 seedlings in three biological replicates for each mangrove strain), grown on ½ MS agar either non-inoculated or inoculated with single bacterial strains(s), were transferred on the top of a submerged ½ MS agar block (disk) to support the seedlings, in which a 24-well plate was filled with 500 μL of liquid media (Appendix A). Control uninoculated seeds (MOCK) were grown in parallel.

For rice (*Oryza sativa* cv. Nipponbare), seeds (12 seeds per treatment) were surface sterilized in 70% Ethanol with 0.05% Tween, rinsed with sterile distilled water five times, and then soaked in 50% sodium hypochlorite solution for 45 min. The seeds were rinsed well with sterile distilled water and transferred aseptically to Phytatray II (Sigma-Aldrich) containing ½ MS + 0.4% Agar, inoculated with or without bacterial strains. The trays were then kept in the dark at 30 °C for 48 h to ensure synchronization in seed germination. The plants were grown for 5 days in growth chambers at 27/25 °C (day/night) with a 12:12 h light/dark cycle, a light intensity of 200 µmol photons m^−2^ s^−1^, and 70% humidity.

For RNASeq analysis, the 5-day-old seedlings (6 seedlings in three replicates for each strain), grown as described above, were transferred to buckets containing Hoagland solution supplemented with 0 or 100 mM NaCl. After 24 h, the shoots and roots were collected separately and flash-frozen in liquid N_2_ before RNA extraction.

### 4.3. Hydroponic Growth Conditions of Oryza sativa cv. Nipponbare

The potential plant growth-promoting (PGP) activity of eight selected bacterial strains was tested hydroponically. As described above, the inoculum was mixed with 50 mL of the ½ MS + 0.4% Agar. On day 5, the seedlings were transferred into 4.3 L buckets filled with half-strength modified Hoagland nutrient solution (5.6 mM NH_4_NO_3_, 0.8 mM MgSO_4_.7H_2_O, 0.8 mM K_2_SO_4_, 0.18 mM FeSO_4_.7H_2_O, 0.18 mM Na_2_EDTA._2_H2O, 1.6 mM CaCl_2_.2H_2_O, 0.8 mM KNO_3_, 0.023 mM H_3_BO_3_, 0.0045 mM MnCl_2_.4H_2_O, 0.0003 mM CuSO_4_.5H_2_O, 0.0015 mM ZnCl_2_, 0.0001 mM Na_2_MoO_4_.2H_2_O and 0.4 mM K_2_HPO_4_.2H_2_O, pH of 5.8) (Appendix A). To avoid contamination and to maintain a continuous nutrient supply, the solution was changed every 3 days. For saline hydroponic growth, the concentration of NaCl was gradually increased until it reached 100 mM NaCl (1st week 0 mM → 2nd week 50 mM → 3rd week 75 mM → 4th week 100 mM). A total of 34 plants were used for each treatment (Appendix A). After 28 days, the plants were scored for their phenotypes.

### 4.4. Greenhouse Experiments

The experiments were conducted in the greenhouse facility at KAUST from February 2022 to September 2023. The 5-day-old rice seedings mock-inoculated or inoculated with either AK164, AK171, or the BiCom (AK164 +AK171, 1:1) were transferred to soil (Stender potting soil mixed 3:1 with rocks) and irrigated with Hoagland solution (0 mM NaCl) or Hoagland solution supplemented with NaCl, gradually increasing the NaCl concentration to 100 mM for the duration of one month (Appendix A). This included 30 plants in three biological replicates.

After the first month, all plants were irrigated with tap water till they were flowering and stopped when tillers were filled with seeds. The panicles were then ripened on the plant until harvest. After 90 days, the plant phenotypic data were collected, including plant height, number of panicles, and spikelets per plant. Two weeks later, the plants were harvested and selected parameters were measured including plant height (cm) and root and shoot DW (g). Other phenotypic parameters included the number of tillers, the number and weight of panicles per plant, the grain yield, as well as the number and weight of grains per plant.

### 4.5. RNA-Seq Library Preparation and Transcriptome Analysis

Total RNA was extracted from roots and shoots of 21-day-old mock-, AK164-, and AK171-colonized plants using Direct-zol RNA Miniprep Plus Kit following the manufacturer’s instructions (ZYMO RESEARCH; Orange, CA, USA) (three biological replicates, each consisting of six roots and six shoots for each strain per treatment). The quality and quantity of the RNA were assessed by Agilent 2100 Bioanalyzer (RNA integrity number >7) and Qubit TM 3.0 Fluorometer (Thermo Fisher, Waltham, MA, USA) with TNA BR assay kit. Samples were processed further for library preparation, sequencing, and bioinformatic analysis (Novogene Co., Ltd., Beijing, China). The RNASeq data have been submitted to the public database GEO and were assigned GSE269165. Genes with adjusted *p*-value < 0.01 and |log2 (FoldChange)| > 1 were considered differentially expressed compared to control.

The reference genome *O. sativa* Japonica cv. Nipponbare (IRGSP-1.0) and gene model annotation files were downloaded from RAP-DB [72,73] (https://rapdb.dna.affrc.go.jp/) (accessed on 11 February 2024). Hisat2 v2.05 was used to align the RNAseq reads with the reference genome to generate a database of splice junctions on the gene models. Next, StringTie (v.3.3b ) [74] was used to assemble and quantify full-length transcripts representing multiple splice variants for each gene locus and feature Counts v1.5.0-p3 to count the read numbers mapped to each gene. Then, the FPKM of each gene was calculated based on the gene length and read count mapped to this gene. We then used R for MacOS. Further, the enrichment analyses were performed using cluster Profiler R package (version 4.4.2) with corrected *p*-value cut-off < 0.05 to call out statistical significance. Gene Ontology (GO) enrichment analysis of differentially expressed genes was implemented by the DESeq2, in which gene length bias was corrected. GO terms with a corrected *p-value* less than 0.05 were considered significantly enriched by differential expressed genes. KEGG was used for functions and utilities of the biological system (http://www.genome.jp/kegg/) (accessed on 3 April 2024).

### 4.6. Histochemical Staining of Rice Roots

Roots of 16-day-old rice plantlets were fixed in fixative (Formaldehyde: Ethanol: Acetic Acid, 10%:50%:5%), and the semi-thin sections were cut from the zone 3 cm above the root tip (three roots in three biological replicates per either strain or BiCom per treatment). Briefly, the root samples were embedded in 5% low-melting agarose, and 100-um-thick sections were cut with the help of a vibratome (Leica VT1200S, Leica Biosystems, Deer Park, IL, USA). The sections were stained with Calcofluor white, Fluorol Yellow, and Basic Fuchsin before being visualized using the Zeiss LSM880 (Jena, Germany) confocal microscope as described in Sexauer et al. [75].

### 4.7. Statistical Analysis

The data from the plant screening assay were subjected to non-parametric one-way ANOVA and the Kruskal–Wallis test [76]. The statistical difference is based on Dunn’s multiple comparison tests. All statistical analysis was performed using GraphPad Prism version 9.5.0 (525) software (https://graphpad.com) (accessed on 4 April 2024).

## 5. Conclusions

At present, it becomes clear that PGPB strains have highly conserved mechanisms for interacting with plants to enhance stress resilience. The underlying mechanisms slowly emerge and suggest that many beneficial mechanisms are based on reprogramming the plant genomes. Importantly, the beneficial microbes not only enhance stress tolerance but also the growth and yield of various crops. Our data unanimously show that two mangrove strains, *Isoptericola* AK164 and *Tritonibacter mobilis* AK171, along with their BiCom, can enhance the resilience of monocot and dicot plants to waterlogging and salinity stress without any growth and yield penalty. Overall, our results highlight the potential of symbiotic bacteria as an eco-friendly innovation towards sustainable agriculture in the face of the challenges of climate change.

## Figures and Tables

**Figure 1 ijms-26-09317-f001:**
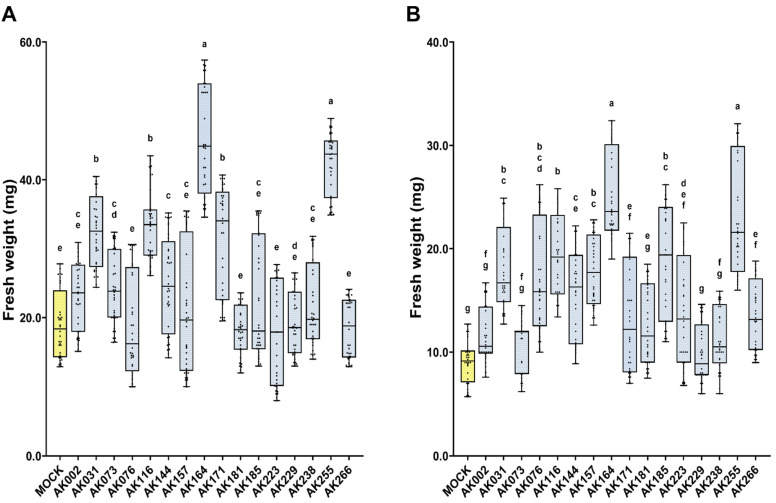
**Mangrove strains enhance the growth of *A. thaliana* under waterlogging conditions.** Growth enhancement of *A. thaliana* seedlings inoculated with candidate bacteria compared to non-inoculated seedlings (MOCK, yellow) in the waterlogging method. Box plots represent FW of *A. thaliana* grown on (**A**)—½MS, (**B**)—½MS + 100 mM NaCl. The statistically significant differences based on one-way ANOVA GraphPadPrism followed by post hoc Tukey’s analysis are represented as a compact letter display of all comparisons (*p* < 0.05).

**Figure 2 ijms-26-09317-f002:**
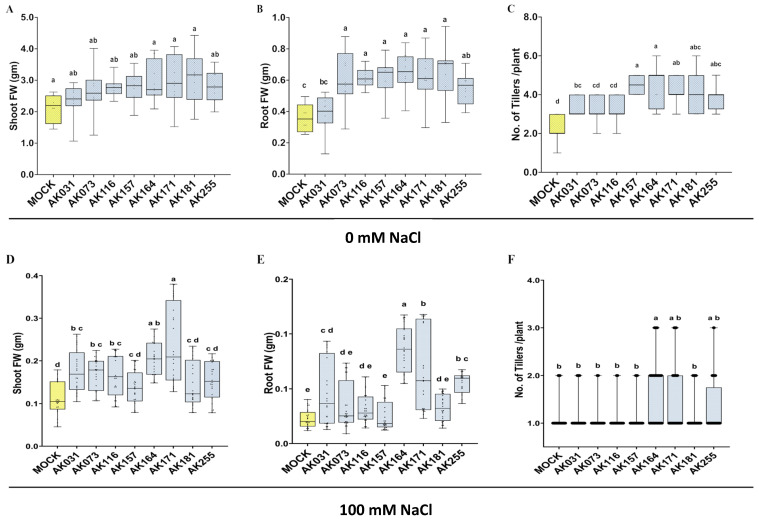
**Mangrove bacteria enhance the growth of rice.** Hydroponically cultivated rice *O. sativa* cv. Nipponbare was analyzed at 28 days post-inoculation with different bacterial strains or mock-inoculation (yellow) at either 0 mM NaCl (**A**–**C**) or 100 mM NaCl (**D**–**F**). (**A**,**D**)—shoot fresh weight, (**B**,**E**)—root fresh weight, (**C**,**F**)—number of tillers per plant. The statistically significant differences based on ANOVA followed by post hoc Tukey’s analysis are represented as a compact letter display of all comparisons (*p* < 0.05).

**Figure 3 ijms-26-09317-f003:**
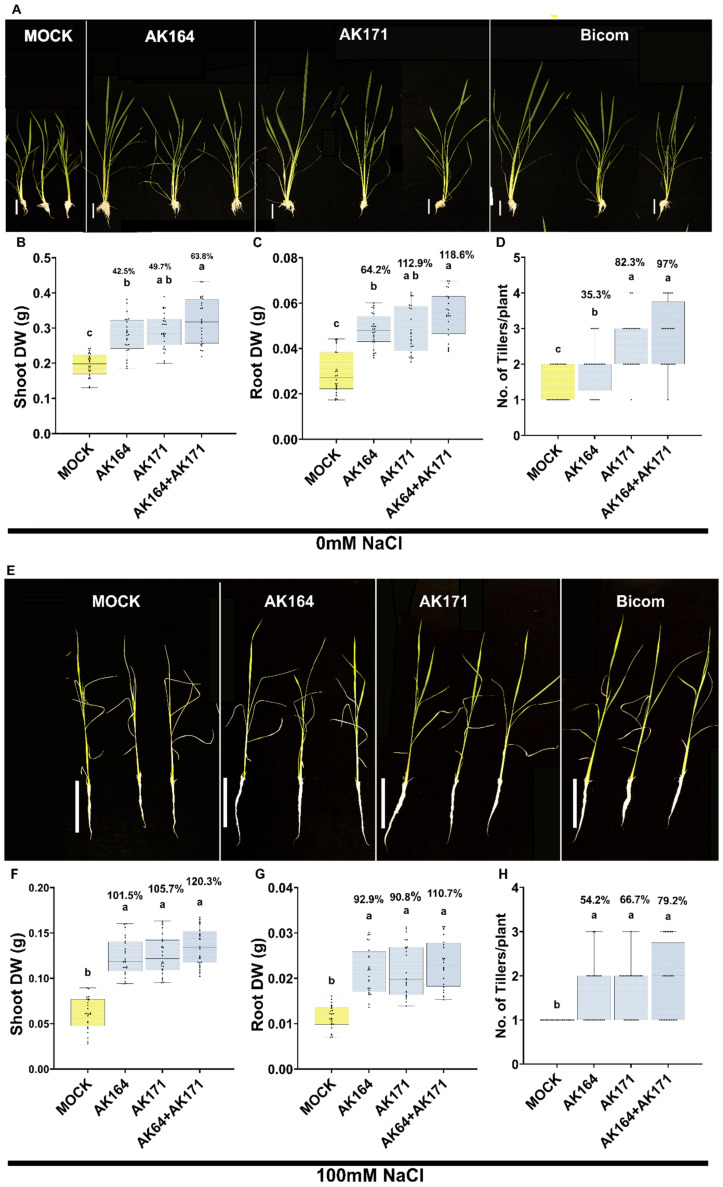
**AK164, AK171, and BiCom enhance rice growth under hydroponic normal and hydroponic saline conditions.** (**A**,**E**)—Plant phenotypes of hydroponically or hydroponically saline-grown rice at 28 days post-inoculation with MOCK (yellow), AK164, AK171, or BiCom. Bar = 10 cm. Percent beneficial increase in growth and number of tillers under hydroponic normal or hydroponic saline conditions of inoculated compared to non-inoculated (MOCK) plants: (**B**,**F**)—shoot dry weight (g), (**C**,**G**)—root dry weight (g), (**D**,**H**)—number of tillers per plant. The statistically significant differences based on ANOVA followed by post hoc Tukey’s analysis are represented as a compact letter display of all comparisons (*p* < 0.05).

**Figure 4 ijms-26-09317-f004:**
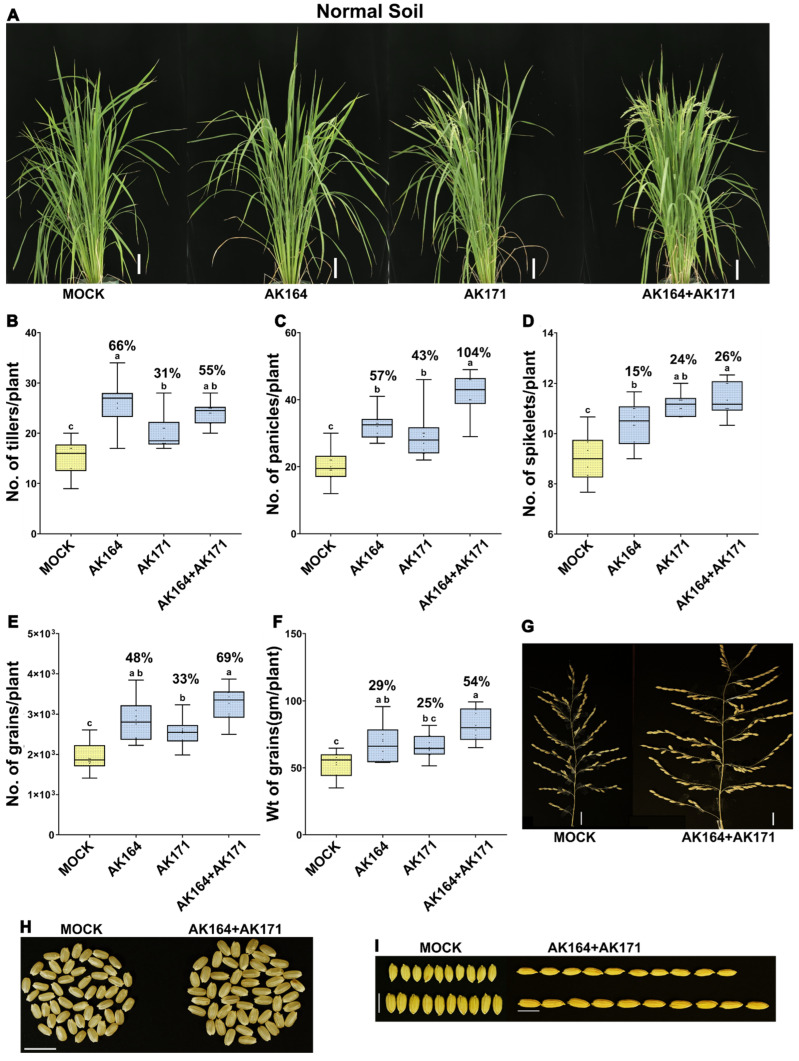
**AK164, AK171, and BiCom enhance the growth and yield of soil-grown rice.** Growth of *O. sativa* cv. Nipponbare inoculated with AK164, AK171, BiCom (AK164 + AK171), or MOCK (yellow) on soil (0 mM NaCl). (**A**)—Rice phenotypes at 90 days of growth (Bar = 10 cm). (**B**–**F**) Percent beneficial increase in the number of tillers per plant (**B**), number of panicles per plant (**C**), number of spikelets per panicle (**D**), number of grains per plant (**E**), weight of grains per plant (**F**), comparison of panicle size of mock and BiCom-inoculated rice (Bar = 2 cm) (**G**). (**H**) Seed size (Bar = 10 mm) and (**I**) seed length (Bar = 7 mm). The statistically significant differences based on ANOVA followed by post hoc Tukey’s analysis are represented as a compact letter display of all comparisons (*p* < 0.05).

**Figure 5 ijms-26-09317-f005:**
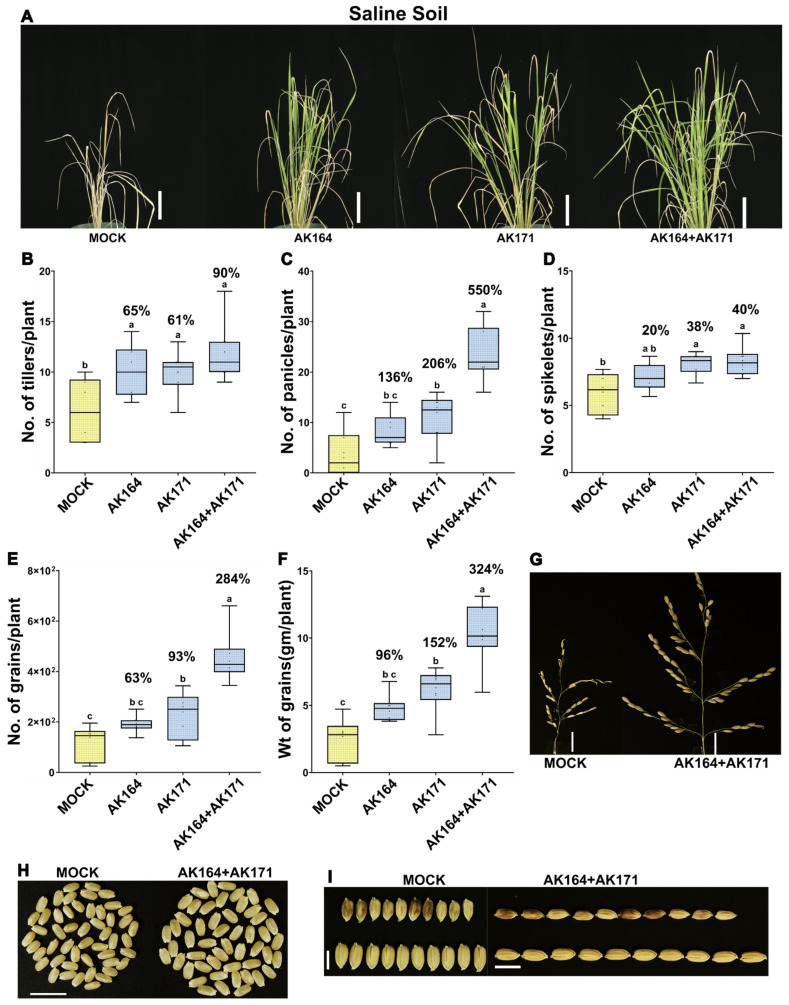
**AK164, AK171, and BiCom enhance the growth and yield of saline soil-grown rice.** Growth of *O. sativa* cv. Nipponbare inoculated with AK164, AK171, BiCom (AK164 + AK171), or MOCK (yellow) on saline soil (100 mM NaCl). (**A**)—Rice phenotypes at 90 days of growth (Bar = 10 cm). (**B**–**F**) Percent beneficial increase in the number of tillers per plant (**B**), number of panicles per plant (**C**), number of spikelets per panicle (**D**), number of grains per plant (**E**), weight of grains per plant (**F**), comparison of panicle size of mock and BiCom inoculated rice (Bar = 2 cm) (**G**). (**H**) seed size (Bar = 10 mm) and (**I**) seed length (Bar = 7 mm). The statistically significant differences based on ANOVA followed by post hoc Tukey’s analysis are represented as a compact letter display of all comparisons (*p* < 0.05).

**Figure 6 ijms-26-09317-f006:**
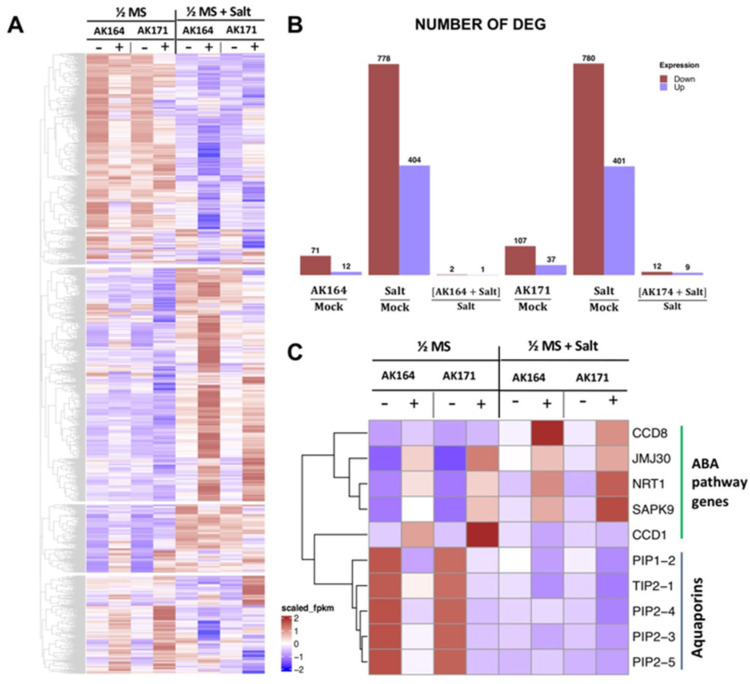
**Heatmap of RNA sequencing data.** DEGs of rice plants grown under normal and saline hydroponic stress conditions treated with AK164, AK171, or MOCK-inoculated. (**A**)—Overview of DEGs in rice roots colonized with AK164, AK171, or MOCK under normal (0 mM NaCl) and saline (100 mM NaCl) hydroponic conditions. (**B**)—Number of DEGs in different conditions. (**C**)—Group of selected root DEGs (log2 > 1, *p* < 0.01).

**Figure 7 ijms-26-09317-f007:**
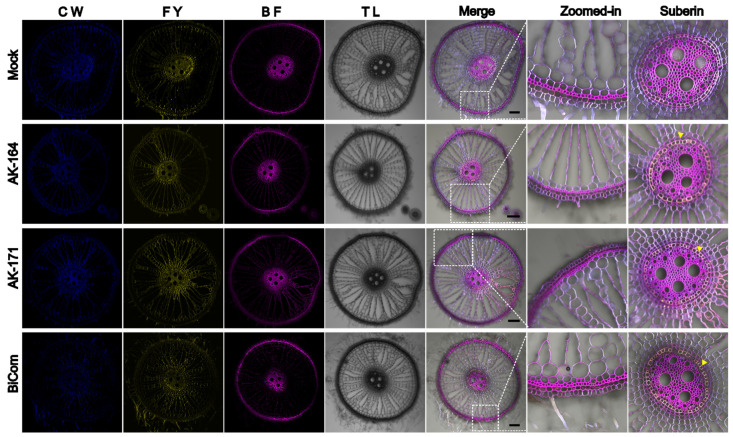
**AK164, AK171, and BiCom enhance lignin, suberin, and sclerenchyma formation in rice roots.** Root sections of plants colonized with MOCK, AK164, AK171, or BiCom (AK164 + AK171) were grown under hydroponic conditions. Multi-serrate sclerenchyma (Zoomed images) in AK164 + AK171 roots and enhanced suberin deposition in the endodermal layer (last panel, yellow arrowheads indicate suberin deposition). Panels (left to right) show Calcofluor white (CW), Fluorol Yellow (FY), Basic Fuchsin (BF), transmission light (TL), and merged channel images. Changes in the sclerenchyma structure are shown in the zoomed-in panel. Bar = 100 µm.

**Figure 8 ijms-26-09317-f008:**
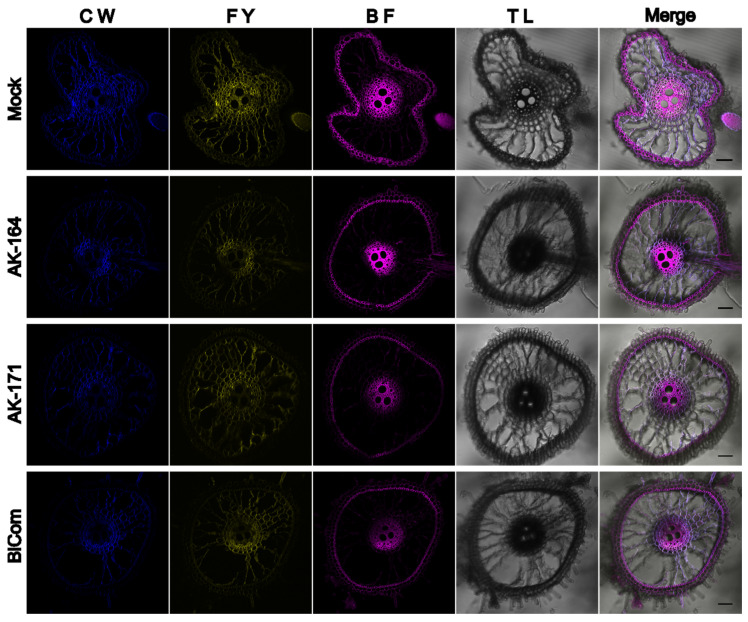
**Collapse of rice roots under hydroponic saline conditions is avoided in AK164, AK171, and BiCom colonized plants.** Root sections of *O. sativa* cv. Nipponbare colonized with Mock shows structural collapse that is not seen in AK164, AK171, or BiCom colonized plants. Triple staining was performed with Calcofluor white (CW), Fluorol Yellow (FY), and Basic Fuchsin (BF). Transmission light (TL) and merged channel images are shown as indicated. Scale bar = 50 µm.

**Table 1 ijms-26-09317-t001:** Beneficial increase (%) in *A. thaliana* fresh weight (g) by 16 candidate strains under submergent growth conditions. * *p* > 0.05, ** *p* > 0.01, *** *p* > 0.001, **** *p* > 0.0001.

No.	Strain	Name	½ MS	½ MS +100 mM NaCl
1	AK002	*Paenibacillus lautus* (OR447938)	9.5	21.8
2	AK031	*Bacillus seohaeanensis* (OR447774)	45.8 ****	96.1 ***
3	AK073	*Thalassospira tepidiphila* (OR447979)	12.8	24.4
4	AK076	*Tritonibacter mobilis* (OR447785)	−8.2	87.3 ****
5	AK116	*Halobacillus locisalis* (OR447813)	43.0 ***	125.2 ***
6	AK144	*Microbulbifer elongatus* (OR447909)	17.5 *	59.7 **
7	AK157	*Demequina activiva* (OR447779)	−1.9	78.5 ****
8	AK164	*Isoptericola chiayiensis* (OR447867)	107.0 ****	165.3 ****
9	AK171	*T. mobilis* (OR447787)	35.2 *	47.3 *
10	AK181	*Pseudomonas azotoformans* (OR447934)	−20.7	57.4 *
11	AK185	*T. mobilis* (OR447793)	1.6	102.5 ****
12	AK223	*P. azotoformans* (OR447928)	−10.1	58.0 *
13	AK229	*Martelella mangrovi* (OR447916)	−3.1	11.2
14	AK238	*Celerinatantimonas diazotrophica* (OR447776)	6.9	17.9
15	AK255	*I. chiayiensis* (OR447854)	101.4 ****	146.1 ****
16	AK266	*P. lautus* (OR448017)	9.5	48.7 **

## Data Availability

The original data presented in the study are openly available in NCBI at https://www.ncbi.nlm.nih.gov/geo/query/acc.cgi?acc=GSM8307357 (accessed on 2 December 2024). The following secure token has been created to allow review of the transcriptome data: mhgjmcqcvfkhpch.

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
