# Peer review of "Mangrove-Derived Endophytic Bacteria Enhance Growth, Yield, and Stress Resilience in Rice"

_ijms, 2025, doi:10.3390/ijms26199317_

Round 1

Reviewer 1 Report

Comments and Suggestions for Authors

This study provides the first evidence that mangrove-derived endophytic bacteria can simultaneously enhance rice tolerance to combined waterlogging and salinity stresses while increasing grain yield, underscoring their high potential for field application. However, all performance data were generated under controlled greenhouse/hydroponic conditions;  field trials are missing. Moreover, whole-genome sequences of AK164 and AK171 have not been reported, and mechanisms rely solely on RNA-seq correlations; metabolomic profiles, protein–protein interactions, and bacterial mutant complementation assays are absent. Additionally, multiple comparisons employed Kruskal-Wallis with Dunn’s test; controlling the false-discovery rate (e.g., Benjamini–Hochberg) would strengthen significance claims.  Additional minor points to improve:

  1. Literature gaps in Introduction: Add some recent papers demonstrating PGPB-mediated flood or salt tolerance in cereals/rice (e.g., Zhao et al., 2022; Li et al., 2023). Include 2-3 studies on mangrove PGPB applied to tomato/maize for comparative benchmarking. Insert a concise paragraph summarising known mechanisms (ACC-deaminase, EPS, ROS scavenging) to frame the novelty of the present work.
  2. Methodological transparency: Provide a supplementary table detailing the selection rationale (antagonism tests, compatibility assays) that led to the 1:1 BiCom ratio. State the exact CFU dose-response curve (e.g., 10⁶–10⁸ CFU mL⁻¹) used to set the final inoculum concentration.

Author Response

Reviewer 1:

“However, all performance data were generated under controlled greenhouse/hydroponic conditions;  field trials are missing”

Response: We fully agree that field trials are critical for assessing the performance of PGPRs, however, currently, it is beyond the scope of our study. That said, we tested our microbes under the greenhouse conditions, which is commonly used as a reliable first step in evaluating the performance of microbial strains.

Moreover, whole-genome sequences of AK164 and AK171 have not been reported, and mechanisms rely solely on RNA-seq correlations; metabolomic profiles, protein–protein interactions, and bacterial mutant complementation assays are absent.

Response: We have already published the whole-genome sequences of both AK164 and AK171; and have updated this information in the manuscript (L 136), which we missed providing earlier. In these studies, we conducted in-depth characterizations of both AK164 and AK171, providing the phenotypic and PGP traits along with genomic insights of both these bacterial strains.

Regarding the bacterial mutant complementation, we appreciate the reviewer's insight; however, the current manuscript was focused on in-depth screening of selected natural wild type bacterial strains for their ability to promote resilience in plants and the possible mechanisms. Surely, we will generate mutants and then use complementation studies in the future.

Additionally, multiple comparisons employed Kruskal-Wallis with Dunn’s test; controlling the false-discovery rate (e.g., Benjamini–Hochberg) would strengthen significance claims.

Response: Since we are comparing a single variable, for example plant biomass, between multiple treatments, we carried out a standard statistical pipeline by using one-way ANOVA with post hoc test (tukey’s HSD). Since we are not testing many traits/parameter, we decided not to use Benjamini–Hochberg.

Literature gaps in Introduction: Add some recent papers demonstrating PGPB-mediated flood or salt tolerance in cereals/rice (e.g., Zhao et al., 2022; Li et al., 2023). Include 2-3 studies on mangrove PGPB applied to tomato/maize for comparative benchmarking. Insert a concise paragraph summarising known mechanisms (ACC-deaminase, EPS, ROS scavenging) to frame the novelty of the present work.

Response: We thank reviewer for highlighting the gaps in our introduction. We have updated the Introduction by adding some recent research conducted in PGPB isolated from mangrove plants, and also have added a short paragraph summarizing the mechanisms used by PGPRs in modulating stress tolerance in plants.

The following information has been added to the Introduction section (L 54).

“Recent studies increasingly demonstrate that plant growth-promoting bacteria (PGPB) isolated from halophytes, such as mangroves, can enhance stress tolerance in plants. Yang et al. [1] investigated the diversity and functional roles of phyllosphere microbial communities in mangrove plants and reported that two strains, Pantoea stewartii A and Bacillus marisflavi Y25, improved salt tolerance in rice by modulating osmolytes, including sugars and proline, and regulating salt-stress-related genes such as OsSOS1, OsPIN1, and OsCIPK15. Similarly, the endophytic bacterium Delftia tsuruhatensis DYX29, isolated from Kandelia obovata, enhanced rice growth under salt stress by promoting osmoregulation, increasing antioxidant enzyme activity, maintaining ion homeostasis, and modulating phytohormone levels [2].

PGPRs enhance plant stress resilience through multiple mechanisms. One such mechanism is through the production of exopolysaccharides, which  helps plants by improving soil aggregation, water retention, nutrient availability, root adhesion and biofilm formation; and stress tolerance by sequestering toxic ions under saline conditions [3-5]. PGPRs also modulate the phytohormone levels to alleviate stress in plants. The ACC deaminase-producing PGPRs break down ACC, an ethylene precursor, reducing stress-induced ethylene levels and mitigating growth inhibition [6, 7]. Ethylene also regulates ROS accumulation under salt stress via the receptor-like kinase SIT1[8]. By lowering ethylene, ACC deaminase-active PGPR increase antioxidant enzyme activities (SOD, CAT, POD), protecting cells from oxidative damage [5]. These mechanisms underscore the interplay between ethylene and ROS and how PGPR alleviate stress to improve plant performance.

Provide a supplementary table detailing the selection rationale (antagonism tests, compatibility assays) that led to the 1:1 BiCom ratio

Response: For our study, we selected the two best-performing bacterial strains and mixed them in a 1:1 ratio (after uniformly adjusting their OD) for testing their ability to promote stress tolerance as a Bicom. We followed the standard protocol for generating synthetic communities, which is widely acknowledged in the field. The aim of our study was not to look at one-one interaction between AK164 and AK171 but to test their combined performance in alleviating stress on plants.

Reviewer 2 Report

Comments and Suggestions for Authors

Dear Authors!

The manuscript is clear, relevant for the field and presented in a well-structured manner.

The manuscript is scientifically sound and the experimental design is appropriate to test the hypothesis.

The manuscript’s results are reproducible based on the details given in the methods section.

The figures/tables/images are appropriate. They properly show the data. The data is interpreted appropriately and consistently throughout the manuscript. I did not find mistakes in statistical analysis or data acquired from specific databases.

The conclusions are consistent with the evidence and arguments presented.

My comments:

Line 59. What stress did you use to enhance the resilience of O. sativa cv. Nipponbare? Add this information.

Table 1, Figures. You should change gm to g.

Figures 4 I, 5I. You should add explanatory inscriptions of group 1 and 2 seeds explaining how their plants were treated.

Lines 571, 619, 620. You should delete numbers 11, 35, 36. Check the design of the references. The cited references are mostly (70%) out of the last 5 years, but I think they are relevant and does not include much self-citations.

Line 257. You should correct the sentence  “…prevent diffusion through plasma membranes.” For example to “… prevent water diffusion out of root vascular cylinder”.

Line 259. You should correct “…AK164, AK171, or BiCom…” to “…AK164-, AK171-, or BiCom-treated …”.

Lines 360-362. Probably, you should rephrase the sentence to “In contrast, the ostps8 mutant showed significantly reduced soluble sugars accumulation, Casparian bands development due to decrease of suberin deposition in the roots compared to the WT and overexpression lines”.

Line 403. Did overexpression of OsPIP2;1 decrease root length, etc? You should add the word  overexpression.

Methods. You should add information about time of seed inoculation with PGPB.

Comments on the Quality of English Language

Line 395. You should check the English of the manuscript. For example, correct suggest to suggests.
